# Quantitative investigation reveals distinct phases in Drosophila sleep

Xiaochan Xu[1,5], Wei Yang[2,5], Binghui Tian[3], Xiuwen Sui[3], Weilai Chi[3], Yi Rao[1,2] & Chao Tang [1,3,4✉]

The fruit fly, *Drosophila melanogaster*, has been used as a model organism for the molecular and genetic dissection of sleeping behaviors. However, most previous studies were based on qualitative or semi-quantitative characterizations. Here we quantified sleep in flies. We set up an assay to continuously track the activity of flies using infrared camera, which monitored the movement of tens of flies simultaneously with high spatial and temporal resolution. We obtained accurate statistics regarding the rest and sleep patterns of single flies. Analysis of our data has revealed a general pattern of rest and sleep: the rest statistics obeyed a power law distribution and the sleep statistics obeyed an exponential distribution. Thus, a resting fly would start to move again with a probability that decreased with the time it has rested, whereas a sleeping fly would wake up with a probability independent of how long it had slept. Resting transits to sleeping at time scales of minutes. Our method allows quantitative investigations of resting and sleeping behaviors and our results provide insights for mechanisms of falling into and waking up from sleep.

[1] Peking-Tsinghua Center for Life Sciences, Academy for Advanced Interdisciplinary Studies, Peking University, Beijing, China. [2] Capital Medical University, School of Life Sciences, Peking University, PKU-IDG/McGovern Institute for Brain Research, and Chinese Institute for Brain Research, Beijing, China. [3] Center for Quantitative Biology, Academy for Advanced Interdisciplinary Studies, Peking University, Beijing, China. [4] School of Physics, Peking University, Beijing, China. [5] These authors contributed equally: Xiaochan Xu, Wei Yang. ✉email: tangc@pku.edu.cn

Sleep is a universal physiological state among species. It plays an irreplaceable role in many aspects of life, ranging from regulating the body's metabolism[1–3] and immunity[4,5], improving learning and memory[6,7] to cleaning up the brain[8]. Sleep disorder is associated with neurodegenerative diseases[9,10] and an increased risk of cardiovascular disease[11,12]. As a simple yet powerful model system, the study of fruit fly sleep behavior has led to the discoveries of important genes and mechanisms, which are also conserved in mammals[13–15]. The flies' sleep behavior is usually characterized as a long-term resting behavior[16]. The animals hardly respond to moderate stimuli in a prolonged resting period, and they perform sleep rebound after sleep deprivation[17]. Before flies fall asleep, there is a latent phase in which flies display the same rest posture as they sleep but reduce their consciousness gradually. Thus, it is hard to distinguish whether a fly is in the sleep latency phase or sleep phase with undisturbed observation. On the other hand, judging flies' sleep state by disturbing flies' sleep with stimuli is incompatible with measuring sleep time. To make some compromise, previous studies used 5 min as a threshold to extract long-term rest episodes as sleep bouts based on observing the wildtype flies' response to stimuli[17]. However, a unique threshold for all flies is questionable, especially for studies including mutants that are supposed to change their sleep behavior either in the latency phase or sleep phase. For example, this standard will confound mutants with shorter latency and mutants with shorter sleep time[18]. This issue greatly hindered sleep behavior research in flies. The field is calling for accurate and quantitative methods to characterize the sleep behavior of flies.

The short-term rest behavior of flies is fractal and embedded within active ones. Several studies suggest that the temporal structure of the behavior is scale invariant[19,20]. A power law exponent is used to describe the individual fly's short-term rest behavior and the exponent can be tuned by dopamine[21]. But for the long-term rest behavior associated with the sleep phase, the power law is not an accurate characterization of the temporal structure[20]. The long-term rest samples were usually ignored or simply regarded as a tail of the power law distribution, often due to the lack of statistics compared with that of the short rest samples.

In order to develop a systematic framework of quantitative characterization of flies' sleep behavior including both sleep latency phase and sleep phase, we established a device to monitor fly behavior continuously with high spatiotemporal resolution. Our datasets enriched the long rest behavior of flies. Our work demonstrated that the temporal structure of rest behavior included different phases, clearly showing that the state of the fly in long-term rest was distinct from that of short-term rest. The long-term rest was memoryless of time, resulting in an exponential distribution of sleep duration. Including the power law distribution in the latent phase, we concluded a generic sleep pattern of the fruit flies. This pattern described an intrinsic law of the whole sleep process and gave parameters to characterize single fly's sleep properties quantitatively. The parameters of the sleep pattern accurately captured the sleep architecture at different ages and in different environments, as demonstrated by the aging-associated and light-associated changes. Thus, the pattern provides a quantitative method for comparison between different scenarios, as well as insights into the mechanisms of sleep.

## Results

### High resolution tracking platform for fly behavior.
We used an infrared camera tracking method instead of the traditional device (*Drosophila* Activity Monitoring System, DAMS) to monitor the locomotive trajectories of flies. It improved the accuracy of determining the rest time and sleep behavior. It monitored the flies day and night continuously. Our experimental setup included infrared lighting, computer, and observing chambers (Fig. 1a). Two rows of infrared lamps were placed on both sides of the observation area, and the camera was fixed above it for overhead shooting. During an experiment, each of the prepared fruit flies was transferred to a single small chamber, one end of which contained food, while the other was closed with a sponge to prevent flies from escaping and to maintain air circulation. The food filled the whole vertical space, and the flies could not walk on it. The flies could walk but not fly in the empty space of the chambers freely. The whole device was placed in an incubator (25°C, 60% humidity) with alternating 12 h-light and 12 h-dark conditions (referred to LD hereafter). The experiments took 4 days in total. The shooting was started after the flies had been in the chambers for 1 day to stabilize their behavior. The frame rate of the recording was 25 fps, the resolution was 560 × 960, and the shooting lasted for 3 days.

To efficiently track the activity, we designed an image-processing program that automatically tracks the flies in the captured video (Fig. 1b, Supplementary Fig. 1a–d). Flies were detected by tracking the difference between the instant image and the background image. The flies were much brighter than the background (black) under the camera, improving the detection accuracy. Our program could detect the flies even when they walked along the edges of the chambers (Supplementary Fig. 1a, middle row). After trajectory reconstruction, the rest bouts of the fly were determined (see "Methods"). We have two sizes of the observing chambers, larger elliptic ones (Supplementary Fig. 1a) and small tubular ones (Supplementary Fig. 1b). The elliptic chamber provided a 2-dimensional space for the fly to explore the environment (referred to as 2D chamber hereafter), while the tubular chamber limited the fly to move mainly along one dimension (referred to as 1D chamber hereafter).

We validated our tracking platform and the rest behavior measurement by comparing the circadian activities and sleep profiles of the flies in the 1D chamber with the DAMS methods. We virtually separated the tubular chamber into two parts with the midline of the y-coordinate (Fig. 1c). The midline worked like an imaginary infrared beam. We could obtain the activity and sleep profile of the fly by counting how many times the fly crossed the midline, mimicking the measurement done with DAMS.

The flies in the experiments had robust circadian activities. Both male flies (Fig. 1d) and female flies (Fig. 1e) gradually increased their locomotion activity before the light was turned on (ZT0) and turned off (ZT12), known as the anticipation phenomenon of flies' circadian rhythm[22]. The sleep profile of the flies not only showed the difference between light phase and dark phase within the same gender but also the difference between male flies and female flies (Fig. 1f). Both male and female flies slept less in the day, especially females, which is consistent with the experimental results performed in DAMS[23,24].

Previous studies also found that flies have location preference for near a food source during sleep[16,25]. We found the long-term stop locations of the flies distributed closely to the food sites in the experiments in the 2D chamber (Supplementary Fig. 1e–g). Intriguingly, the female flies in the 1D chambers preferred to have siesta at both ends of the tubes in the day, while male flies only preferred the ends with food (Supplementary Fig. 1h–i).

### Quantitative sleep pattern of *Drosophila melanogaster*.
Although sleep is highly variable among individuals and the same fly may behave differently in different environments, a general pattern for the probability distribution of fly rest duration can be determined.

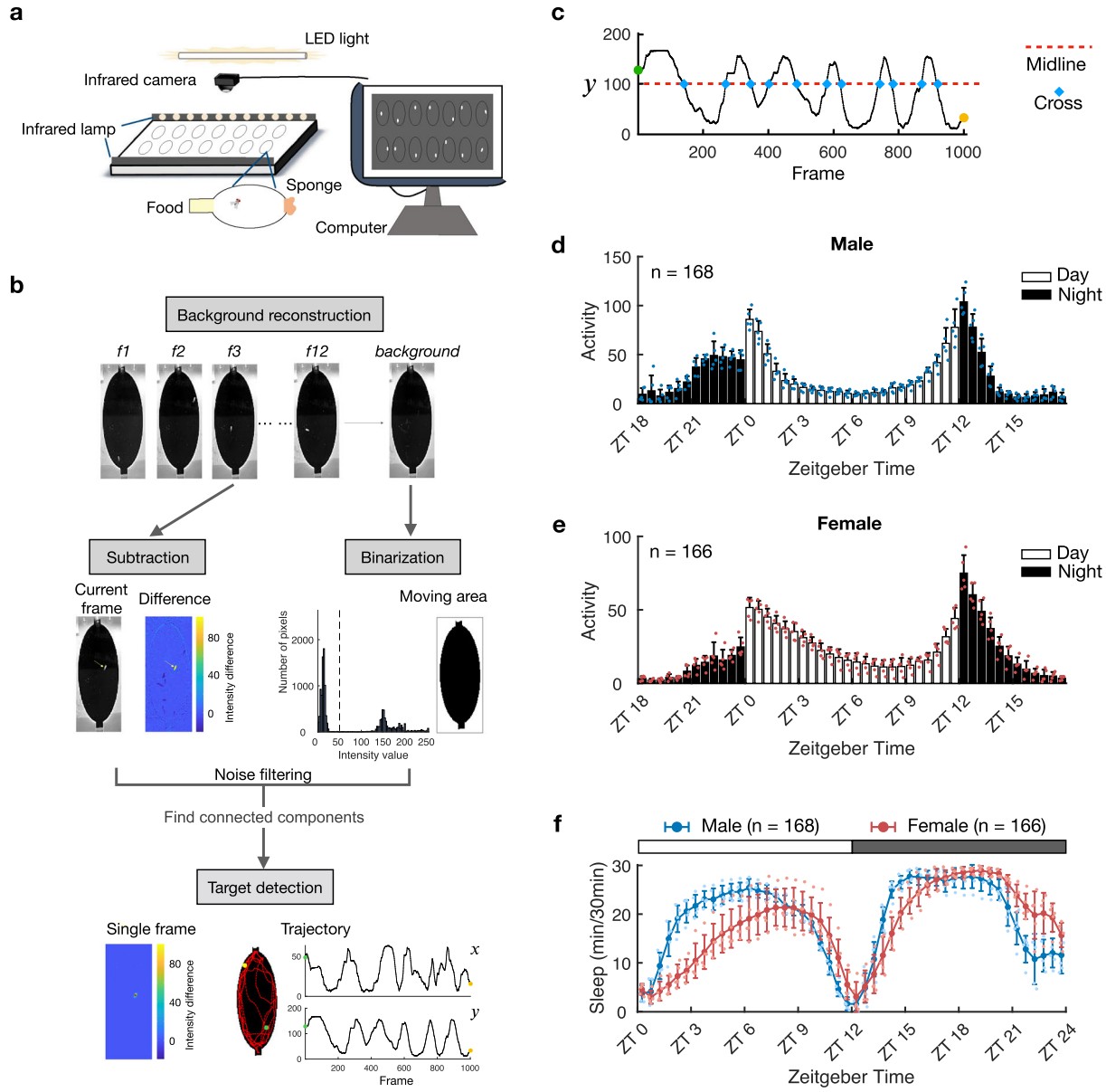

**Fig. 1 High resolution video platform for tracing flies' locomotion activities. a** Schematic of the experimental device. The cold light source was used to provide periodic light to the fruit fly to maintain the normal circadian clock; the infrared camera was placed directly above the observation area; two rows of infrared lights were arranged on both sides of this area; the food was placed at one end of the chamber and a sponge at the other. **b** The videos obtained by the infrared camera were analyzed by computer program (see "Methods"). **c** Imitating the DAMS with the trajectory of the fly. The midpoint of the y-coordinate is used to calculate how many times the fly has interrupted the infrared beam if it is in the DAMS. **d**–**e** Average daily activity profiles of male and female flies in 3 days (72 h) within each 30 min bin. The white and black columns indicate activities during the light and dark phases, respectively. **f** Average sleep profiles of male and female flies in 3 days within each 30 min bin. The white and black bars above indicate the light and dark phases, respectively. The blue curve is for male flies, red curve is for female flies. The error bars in (**d**–**f**) indicate SEM for five biologically independent experiments and n for the number of flies used.

To obtain the temporal distribution of rest behavior, we first classified the rest intervals by the circadian phases. The rest events in the light phase were labelled with "day", and those in the dark phase were labelled with "night".

In each phase and each gender, the distribution of the individual fly's resting time showed a power law decay with a cutoff of around 100 s. It could be seen clearly when the data was presented with the complementary cumulative probability (referred to "probability" hereafter) (Fig. 2a–d).

Let $P(t) = P(X>t)$, where X is the length of the rest time. The probability function $P(t)$ represents the probability of rest bouts

with intervals larger than $t$. Due to the power law decay when $t$ was small, the probability function had a linear relationship on double logarithmic plot (Fig. 2a–d, left panels). This linear relationship lasted from the beginning to the order of hundreds of seconds, spanning two orders of magnitude. As $t$ became larger, the frequency of longer rests deviated from the linear relationship, making the tail of the data curve. The data on this "curved tail" was linear in $t$ on a semi-logarithmic plot (Fig. 2a–d, right panels), indicating an exponential decay on the tail of the distribution. We fitted the tail part of the same data with gaussian decay, which is much faster than an exponential decay

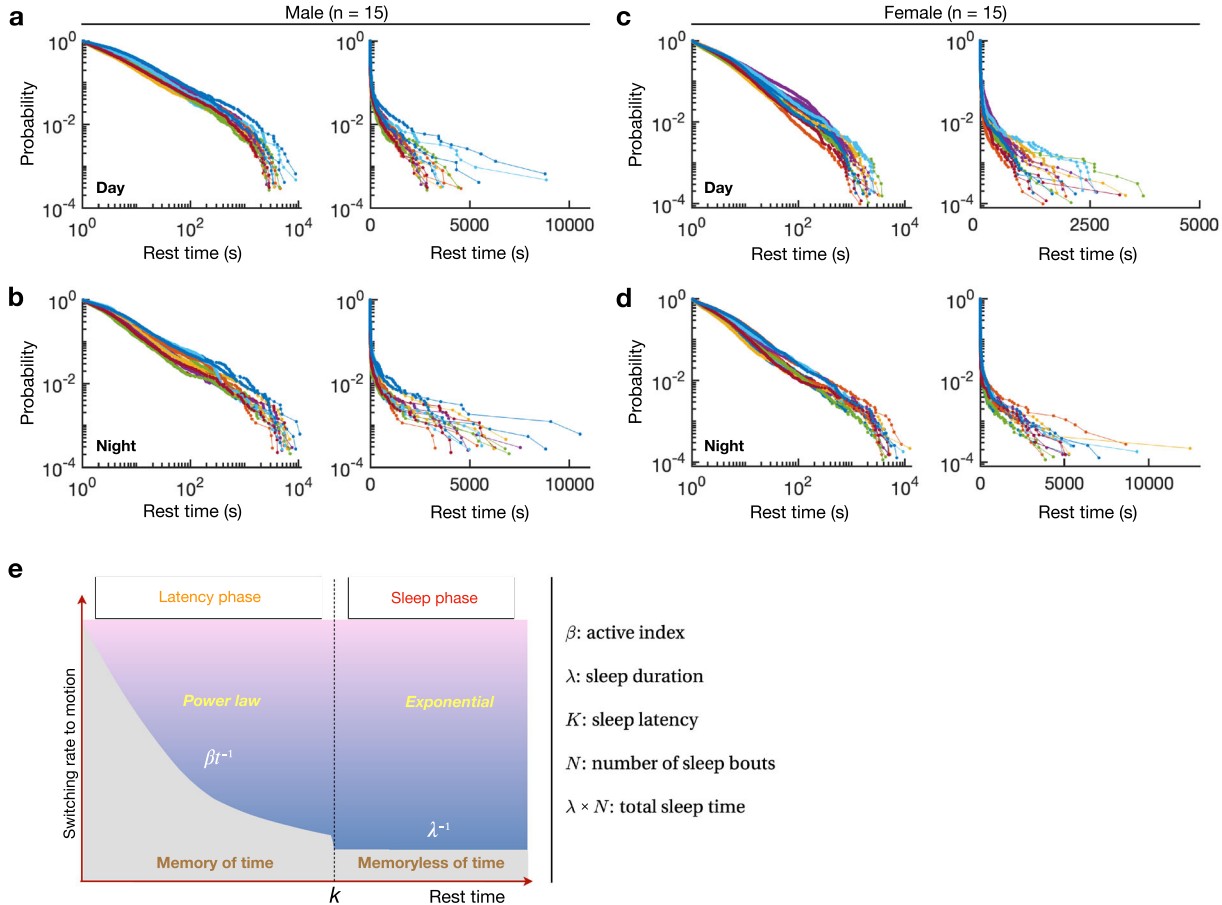

**Fig. 2 Quantitative sleep pattern definition based on the probability distribution of rest-bout duration. a–b** Probability of rest time of male flies in light and dark phases, respectively. **c–d** Probability of rest time of female flies in light and dark phases, respectively. For both the males and females, 15 examples are shown. Graphs on the left panels of (**a–d**) use logarithmic scales on both the horizontal and vertical axes, and on the right panels only use logarithmic scale on the vertical. **e** Schematic of the two phases of sleep behavior in the fly. Power law decay and exponential decay is separated by the sleep latency (*K*). Five sleep parameters (shown on the right) can be extracted from the sleep pattern to quantitatively analyze the sleep behavior of each individual fly.

(Supplementary Fig. 2a–d). The quantified relative Akaike information criterion (AIC) showed that the likelihood of the gaussian distribution model is much lower compared with the exponential distribution model (Supplementary Fig. 2e).

The above observation revealed that there was a specific time $K$ that divided the whole probability distribution into two parts, such that

$$\ln P(t) = \begin{cases} -\beta \ln t + b_1, & t \leq K \\ -\lambda^{-1} t + b_2, & t > K \end{cases} \quad (1)$$

That is,

$$P(t) \propto \begin{cases} t^{-\beta}, & t \leq K \\ e^{-\lambda^{-1} t}, & t > K \end{cases} \quad (2)$$

Thus, the temporal distribution of rest behavior showed that the shorter bouts obeyed a power law distribution, while the longer ones obeyed an exponential decay distribution, referred to as the exponential distribution.

The meaning and implication of the power law distribution and that of the exponential distribution were completely different. If we denote as $w(t)$ the switching rate from rest to movement at instant moment $t$, we get:

$$dP(t) = -P(t)w(t)dt \quad (3)$$

For the power law distribution, this gives

$$w(t) = \frac{\beta}{t}, \ t \leq K \quad (4)$$

This form of switch rate is universal for power law distributions, that is, after the fly entered the resting state, the rate of switching from rest to motion is inversely proportional to the time it has rested. In other words, the longer a fly stayed at rest, the more likely it would continue to rest, implying that the fly had a memory of the rest duration. Obviously, this state cannot be continued indefinitely, otherwise the fly would never move again. Consistent with this, the duration of rest obeys a power law distribution only below a threshold period $K$, above which the fly entered another phase with an exponential distribution.

For the exponential distribution, the switch rate takes the form

$$w(t) = \lambda^{-1}, \ t > K \quad (5)$$

In this phase, the probability of switching from rest to motion was no longer affected by the duration but was instead equal to a constant. The memory of duration disappeared.

Note that in previous studies of flies, a rest interval ≥ 5 min was used as a criterion for sleep phase. Here we found that there was indeed a state transition in resting behavior at around this time scale. The relatively rare long-term rest behavior corresponding to sleep displays a different statistical characteristic compared with the short-term rest behavior.

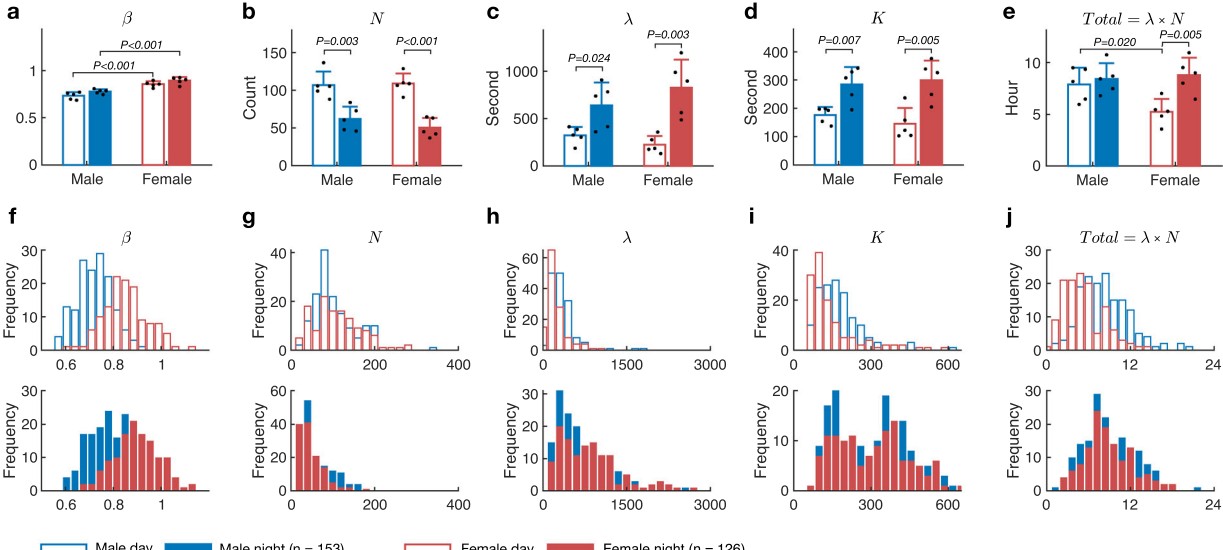

**Fig. 3 Comparison of sleep parameters between the light and dark phases. a** Distribution of $\beta$ (active index). **b** Distribution of $N$ (number of times the fly fell into sleep phase in 3 days). **c** Distribution of $\lambda$ (sleep duration in seconds). **d** Distribution of $K$ (sleep latency). **e** Distribution of total sleep time in 3 days. Each panel of (**a–e**) shows the statistical test (two-sample $t$ test) results of the significance level of difference between male and female and/or in the light phase and dark phase, as indicated. Error bars indicate SEM of five biologically independent experiments. **f–j** The distribution of each index with histograms. Hollow columns represent the light phase, filled columns the dark phase. The blue columns represent male ($n = 153$), and the red columns female ($n = 126$).

In summary, we established a general rule of sleep in individual flies, characterized by the probability of rest duration or equivalently by the switching rate from rest to motion (Fig. 2e). This model consisted of two phases: (1) sleep latency phase: the early stage of rest behavior, characterized by a power law distribution of rest duration. The larger the power law exponent $\beta$, the more probable the fly would move again in a resting state. We called $\beta$ as the *active index*; (2) sleep phase: fly entered a long period of rest, characterized by an exponential distribution of rest duration. The parameter $\lambda$ of the exponential distribution indicated how hard a fly would wake up spontaneously in the sleep phase. This parameter can be used to represent the average *sleep duration* in the sleep phase. From the transition time between the two distinct resting patterns, we got the parameter $K$ as *sleep latency*. With the $K$ threshold, we could count how many times the fly fell asleep ($N$) and estimate the total time (*Total*) the fly spent in the sleep phase ($\lambda \times N$).

**Parameters of the sleep pattern**. Previous studies have observed that the sleep behavior differs between male flies and female flies and between the light phase and dark phase. The sleep pattern we found was prevalent for all flies, but the five parameters $\beta$, $\lambda$, $K$, $N$, and *Total* characterizing the pattern could be different for each individual fly and may vary between day and night, and between male and female (Fig. 3).

In 1D chambers, the active index $\beta$ in flies of the same gender differed slightly between day and night but varied significantly between genders (Fig. 3a). In the light phase, the active index was $0.73 \pm 0.04$ for males and $0.85 \pm 0.03$ for females. In the dark phase, the active index was $0.78 \pm 0.02$ for males and $0.89 \pm 0.04$ for females. Overall, the active index value had a range of $0.7–1$. Female flies' rest time tended to be shorter than male flies. For both genders, the rest time was shorter in the dark phase than the light phase.

Interestingly, unlike the active index, the sleep phase parameters $N$, $\lambda$, and $K$ showed significant differences between the light phase and dark phase rather than between genders (Fig. 3b–d). Compared with the light phase, flies preferred to

sleep longer every time they entered the sleep phase and with a reduced number of sleep bouts in the dark phase (larger $\lambda$ and smaller $N$). Thus, sleep in the light phase is more fragmental. The male had comparable total sleep time for the light phase and dark phase, while the female mainly slept in dark time (Fig. 3e). These differences in the five parameters could also be seen from the distributions of individual flies (Fig. 3f–j).

The sleep latency $K$ was similar between males and females. The $K$ value during the day was around 200 s, and during the night was around 300 s. These values are very close to the 5 min threshold used in transitional studies. Our method provides a more precise and individualized threshold for each fly.

We note that the 1D chamber may be an unnatural exploring context for the flies. To test the generality of the quantitative sleep pattern we found, we performed the same experiments in 2D chambers and compared the results with those in 1D (Supplementary Fig. 3). The flies in 2D chambers had much larger space to explore and could move in different directions. We found that the flies in 2D chambers showed the same probability pattern of the rest time as in 1D chambers (Supplementary Fig. 3a–b). The three parameters ($N$, $\lambda$, and $K$) of the sleep phase varied slightly from 1D to 2D chambers (Supplementary Fig. 3d–f), and the male decreased the sleep latency ($K$) in the dark phase. Notably, the active index in the light phase consistently decreased for both males and females in the 2D chambers (Supplementary Fig. 3c). The change hints that the active index is related to not only light phase but also to the space the flies are exploring.

These results indicate that the general mechanism behind the sleep pattern is universal, but the animal may fine tune the sleep pattern to adapt to the environment, as reflected in the changes of the parameters.

**Age-associated and light-associated changes in sleep pattern**. Sleep is regulated by various internal and external factors. To investigate how these factors may influence the sleep pattern, we chose aging and constant darkness as the representative internal and external factors, respectively. We found that our methods could also quantify the changes associated with these factors.

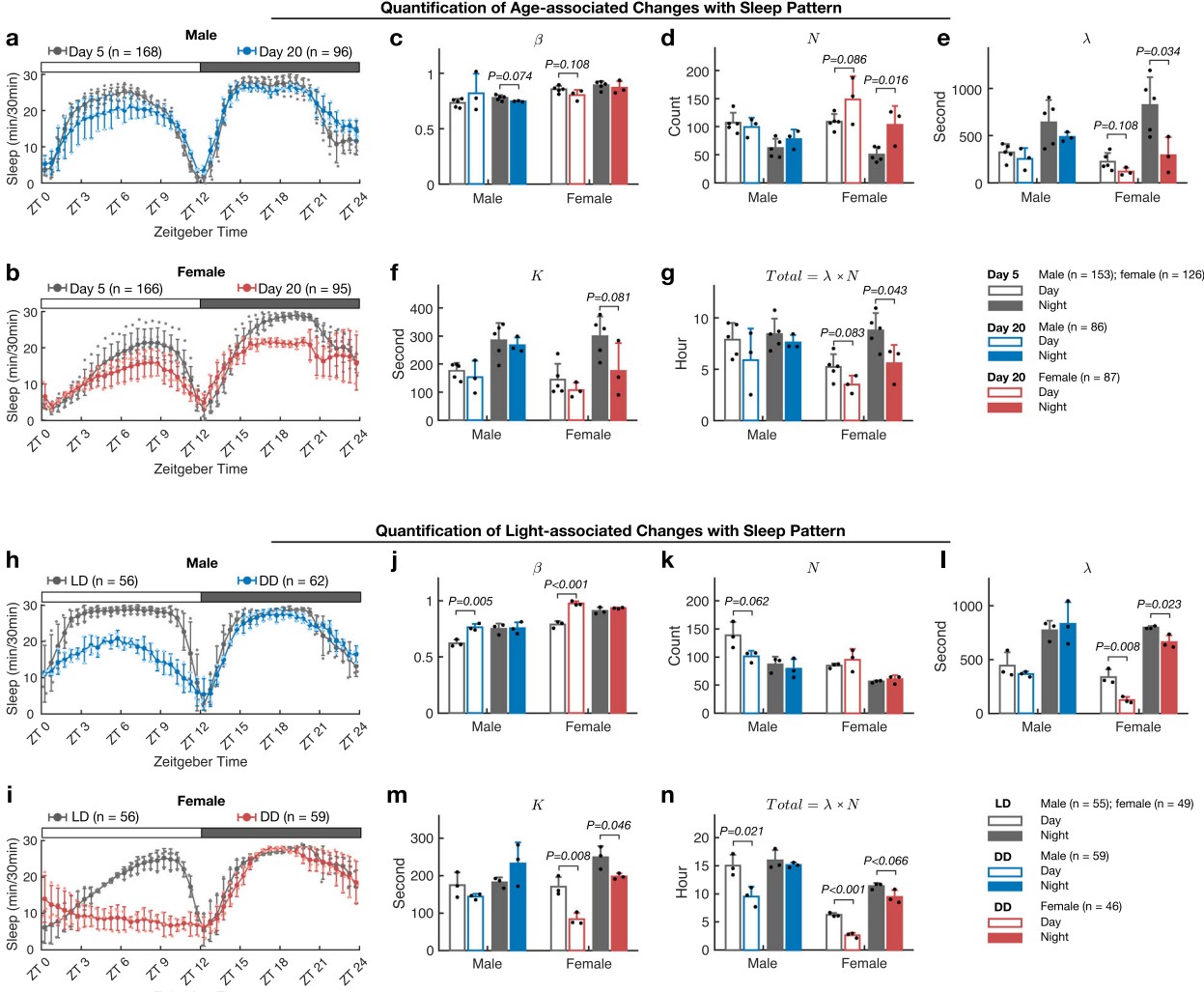

**Fig. 4 Sleep pattern changes in older flies or under constant darkness. a–b** Sleep profiles of older male and female flies (20 days post eclosion) compared with younger flies (5 days post eclosion). The blue, older male; red, older female; grey, younger male or younger female. Error bar indicates SEM of five biologically independent experiments for the younger groups and three biologically independent experiments for the older groups. **c–g** Comparison of the sleep pattern parameters of older flies with the younger ones. The hollow and filled columns represent the mean of the parameter in the light and dark phases, respectively; grey: younger flies; blue, older male; red, older female. **h–i** Sleep profiles of flies under constant dark condition (DD, 12 h:12 h) compared with groups under cyclic light condition (LD, 12 h:12 h). The blue, male in DD; red, female in DD; grey, male or female in LD. Error bar indicates SEM of three biologically independent experiments for the LD groups and three biologically independent experiments for the DD groups. The light phase of DD in **h** and **i** is hypothetically assigned in accordance with the normal light-dark condition (LD, 12 h:12 h). The flies in the DD group had been raised under the LD condition and then transferred to the DD condition. **j–n** Comparison of the sleep pattern parameters of flies under different light conditions. Grey: male or female in LD; blue, male in DD; red, female in DD.

The aging process causes loss of sleep consolidation in elderly human and flies[26,27]. We applied our sleep pattern study to the aging-associated sleep changes. The parameters of the sleep pattern of the older flies (20 days post eclosion) were compared with the young groups (5 days post eclosion).

On day 20, the flies were still normally active (Supplementary Fig. 4a). The older male flies had similar sleep profiles to the younger flies (Fig. 4a), indicating that the sleep architecture is still maintained. This is consistent with the previous observation that the male flies' sleep architecture showed little change until around 3 weeks[27]. Correspondingly, the 5 parameters of sleep did not significantly change in older male flies (Fig. 4c–g).

Unlike the males, the female flies increased their activity dramatically and slept less at day 20 (Fig. 4b, Supplementary Fig. 4a, middle), indicating that their sleep behavior had changed. From the comparison with the younger flies, we found that the

older female fliers' sleep bouts ($N$) increased (Fig. 4d) both in the day and night. On the other hand, the sleep duration parameter $\lambda$ decreased (Fig. 4e). The change in $\lambda$ was very large especially during the night and led to a reduction in older female's total nighttime sleep even though sleep happened more times (Fig. 4g). The fragmental sleep behavior of the female caused by aging was well quantified with our method.

The light cycle is a crucial factor for the circadian and sleep[28]. We then compared the sleep pattern of flies in constant dark condition (DD) and those in normal light-dark condition (LD). The flies transferred to the DD condition increased their activity dramatically (Supplementary Fig. 4b–c). Overall, they slept less in the constant darkness (Fig. 4h, i).

Interestingly, removing light not only changed the sleep profiles but also the active index. In cyclic LD condition, the active index is smaller in the light phase than in the dark phase.

Under constant darkness, this parameter in the light phase significantly increased to a level similar to that of the dark phase (Fig. 4j). (Under DD, the light phase means the period when the light would be on in the LD condition.) The constant darkness had the same effects on the active index of males and females. Thus, the active index may also be regulated by light.

Looking through the sleep profiles (Fig. 4h, i), both the male and female flies remarkably reduced their sleep in the light phase under the DD condition. Comparing the sleep parameters, we found that this reduction of sleep came from different changes in the sleep architecture between the male and female flies. The male flies had fewer sleep bouts (Fig. 4k), the same sleep duration (Fig. 4l), and the same sleep latency (Fig. 4m) when they were in constant darkness. The female flies had the same sleep bouts, shorter sleep duration, and shorter sleep latency. The total sleep time (*Total*), which integrated the changes in sleep bouts ($N$) and sleep duration ($\lambda$), decreased both in male and female flies in the light phase of the DD condition compared with that of LD condition (Fig. 4n), consistent with the observations in Fig. 4h, i.

In the dark phase, the differences between LD and DD conditions were hard to distinguish from the sleep profiles with the standard measurements (Fig. 4h, i), other than that the flies slept slightly more in some hours and less in some other hours. The sleep parameters did not show much difference in male flies either, indicating that the sleep architecture of male flies did not change in the dark phase under DD condition compared with LD condition. However, for females, we observed a decrease in sleep duration ($\lambda$) (Fig. 4l), which led to less total sleep time (Fig. 4n). We also noticed that the sleep latency ($K$) of the females decreased under the DD condition (Fig. 4m). Thus, the female flies fell asleep more quickly and slept a little shorter in DD condition based on our analysis. These subtle changes could hardly be detected by the standard measurements but could be accurately characterized by our method.

Taken together, these results indicate that the sleep pattern is a widely applicable metric to quantify the individual fly's sleep behavior. It can accurately capture the difference when the experimental scenario changes.

**Correlation between sleep parameters**. Noting that the active index tended to change differently from the other sleep parameters, we investigated the correlation between the parameters. We used the rank correlation (Spearman's correlation coefficients) to quantify the correlations. The dataset consisted of all the samples from different settings and conditions discussed before (male/female, light/dark, different ages, different arena sizes, and different light conditions).

Hierarchical clustering classified the ten parameters (five from the light phase, five from the dark phase) into two groups (Fig. 5a). The sleep phase related parameters ($\lambda$, $K$, *Total*) were in one group, and the active index ($\beta$) and sleep bout ($N$) were in the other group. It indicates that the changes in *Total* mainly depended on the changes in sleep duration (Fig. 5b–e). The number of sleep bouts was negatively correlated with the active index $\beta$ (Supplementary Fig. 5), which made sense as a more active fly would fall into sleep less frequently.

The active index was in general negatively correlated with all other parameters in the day. That is probably because both the active index and the other sleep phase parameters are regulated by light. On the other hand, the active index was uncorrelated with the sleep duration, the total sleeping time, and the latency at night (Supplementary Fig. 5). It further suggested that the power law latency phase and the exponential sleep phase were controlled by different biological circuits.

**Mathematical model**. The molecular mechanism leading to the observed statistical properties of flies' sleeping behavior is unknown. Previous efforts were using mathematical models to understand sleep regulation. For example, one of the widely applied models, the "two-process model"[29,30], considered both the circadian and homeostasis regulation of the sleep. The initial intention of that model is different from the theme of this paper, but it reminded us to check if the current sleep length is correlated with the previous sleep history. We found the current sleep duration was not affected by the total amount of previous sleep (Supplementary Fig. 6a–d). In addition, neither the total sleep in the previous day nor night showed correlation with the total sleep in the following night or day (Supplementary Fig. 6e, f). This is probably because the flies in our experiments were under undisturbed conditions and their rest behavior happened spontaneously. Unlike the sleep deprivation experiment, in our experimental settings the flies' sleep pressure could be released anytime. Thus, it is conceivable that in our experiments each rest behavior started approximately with the same state of the fly.

An exponential distribution of time durations (as for the rest time in the sleeping phase) is often associated with a simple two state transition (e.g., from sleep state to wake state) with a memoryless transition rate. On the other hand, a power law distribution of time durations (as for the rest time in the latent phase) is usually more complicated and can be generated by multiple mechanisms. It is known that the human brain has a few different states during sleep[31,32]. Previous research also found that the activity of fly brain has different quiescent states when the fly is resting[33]. It may be conceivable that during one bout of rest, the fly brain changes its state from one to another. Here we present a simple mathematical model based on brain state transitions to account for the observed sleep pattern.

As illustrated in Fig. 6a, the model is a random walk in a space of brain states. Fly has two hemispheres. Assume these hemispheres were identical yet independent, and each hemisphere has $L$ possible states during rest (Fig. 6a). Starting from the origin, the brain state of the fly performs a random walk in the two-dimensional state space of resting. It will continue to rest as long as the brain state is inside the state space. The time interval of a rest bout is the first recurrent time of the random walker back to the boundaries (indicated by the green dots in Fig. 6a). When the walker walks in the 2-D state space, it can only jump to the neighboring states. The dwell time of the walker in a resting state is related to its distance from the origin. Specifically, the dwell time is given by an exponential distribution regarding the switch as a Poisson process, that is:

$$\text{dwell time} \sim e^{-t/\theta}, \theta = (i^2 + j^2)^{\frac{a}{2}} \qquad (6)$$

Furthermore, when the walker is deep inside the resting state space, the brain states turn to the sleep phase (red dots in Fig. 6a). In other words, there is a threshold in the "distance from the origin" (the dash line in Fig. 6a), above which the fly goes from resting to sleeping. Once the brain state enters the sleeping phase, it will jump out (wake up) with a constant rate $\varepsilon$.

The model gave a similar probability curve as the experimental results (Fig. 6b, c). The parameter $\varepsilon$ is closely related to the inverse of the sleep duration ($\lambda^{-1}$) while the parameter $a$ related to the active index $\beta$ (Fig. 6d) and $w$ to the sleep latency $K$ (Fig. 6e). If we restricted the model parameters to be those (marked by asterisks in Fig. 6e) corresponding to experimentally observed latency values (80–300 s), the range of $\beta$ and $K$ values generated by the mathematical model agreed reasonably well with the experimental values (Fig. 6f).

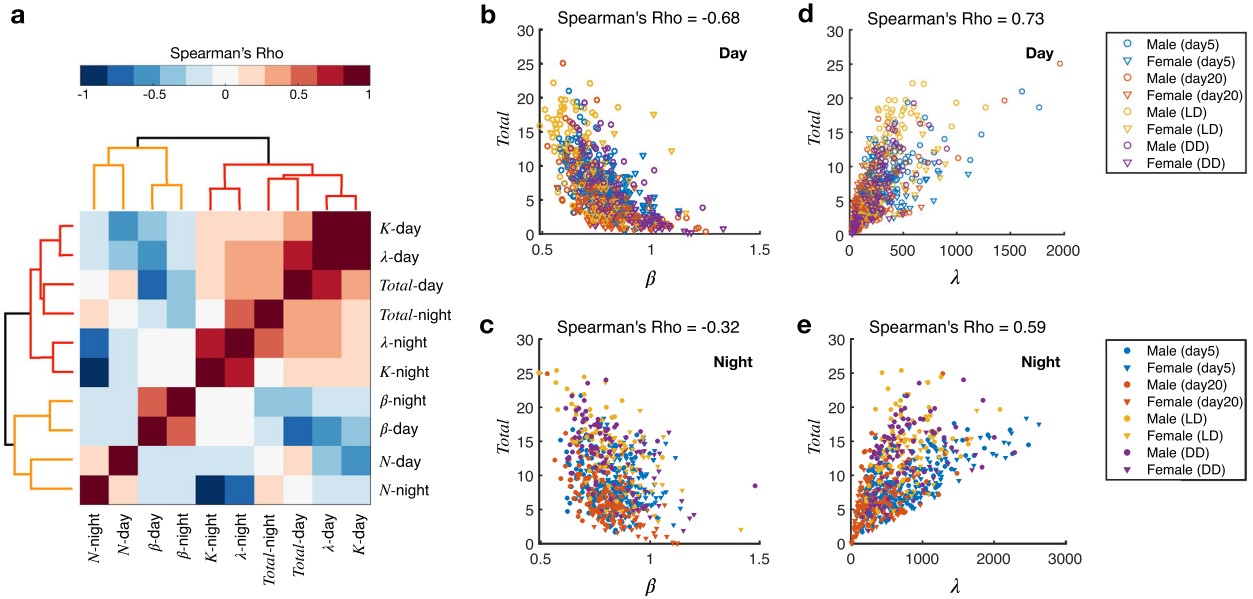

**Fig. 5 Correlation between the parameters of the sleep pattern. a** Heatmap of Spearman's correlation coefficients (Spearman's Rho) of parameter pairs. Each correlation coefficient was calculated with all the flies' data from different experimental conditions. Parameters in the light phase are labeled with the suffix "day" and in the dark phase with the suffix "night". **b–c** Scatter plot of the active index ($\beta$) and total sleep time (in hours) of all the flies in the light phase and the dark phase. **d–e** Scatter plot of the sleep duration ($\lambda$) and total sleep time (in hours) of all the flies in the light phase and the dark phase. In (**b–e**), circle, male; triangle, female; blue, flies 5 days post eclosion in 1D chambers (LD, 12 h:12 h); red, flies 20 days post eclosion in 1D chambers (LD, 12 h:12 h); yellow, flies 5 days post eclosion in 2D chambers (LD, 12 h:12 h); purple, flies 5 days post eclosion in 2D chambers (DD, 12 h:12 h). The correlation coefficients are shown above the scatter plots.

## Discussion

In the past years, power law distributions of rest duration were also observed in mammalian species including mice and humans[34]. In humans, the power law exponent $\beta$ was found to relate to some psychological disorders. The exponent decreases in depression patients ($\beta \approx 0.7$) and bipolar II patients ($\beta \approx 0.7$) compared with a normal value ($\beta \approx 0.9$)[35,36]. In our experiments, the power law exponent for flies was around 0.7–1 (instead of 0.37 as measured previously[19]), which is quite close to the ranges in mice and humans. Furthermore, the sleep duration of mammalians was found to be exponentially distributed by using EEG measurement[37]. Our results in flies suggest that this pattern of rest and sleep can be more universal. Even though the sleep state of flies is short of accurate physiological definition as in mammals, it is still possible to define sleep state quantitatively at the behavior level through the sleep pattern we found. With the five parameters ($\beta$, $\lambda$, $K$, $N$, $Total$), one can obtain quantitative information about each fly's rest and sleep behavior. Given the shared universal pattern with mammals, the fly could be used as a model organism to study rest and sleep behavior.

The Discrete Hidden Markov Model (DHMM) has been used for sleep staging analysis to aid the understanding of sleep behavior in humans[38]. It considers the relationship between the precious sleep state and the next sleep state and captures the properties of stage transition of sleep behavior. It is also a promising model for sleep behavior analysis in flies, and as reported the probability of initiating activity (P(Wake)) and the probability of ceasing activity (P(Doze)) could give a relative measure of sleep[39]. But when applying the DHMM model, those studies hypothesized that the current state is only affected by the previous state in the last time window (short memory). This hypothesis has never been tested biologically and is contrary to the observation that the flies' rest behavior has a power law decay. A power law decay implies that the current state is dependent on how long the animal has been resting, i.e., the state has a memory.

From our analysis, we found that only the sleep behavior has an exponential decay, suggesting that the HHM model with a constant transition rate is probably applicable only in the sleep phase. The sleep pattern we found can also explain why P(Wake) changes along the day and especially showed two peaks around ZT0 and ZT12 in the previous study (see Fig. 1b and c in Wiggin, T. D., et al.[39]). Around those times, the flies move frequently and have mainly shorter rest time, the average level of P(Wake) should then be larger according to P(Wake) ≈ $\beta/t$. Comparing with the previous probabilistic analysis method, the sleep pattern studied here gives a more straightforward framework to quantify the whole sleep process of flies.

We found that the active index ($\beta$) of a fly was correlated with how many times the fly fell asleep ($N$) but not with its sleep duration ($\lambda$). It implied that the two phases of the sleep pattern are associated with two different aspects of sleep: sleep onset and sleep maintenance. It has been found that dopamine signaling regulates the active index[21] and that a variety of genes and neurons are related to sleep behavior[15]. Which circuits can tune the sleep pattern parameters is still an open question.

The observed sleep pattern can be accounted for by a mathematical model of brain state transition. A key feature of the model is that the brain states have different "quietness", and the model predicted that it is more difficult to jump out of a quieter state for the flies. This is consistent with an increased arousal threshold during the fly's sleep latency phase[33,40]. But whether there are indeed such different brain states in *Drosophila* needs more experimental evidence in the future.

Applying our model to analyze the homeostasis is possible but may be challenging since after sleep deprivation the sleep architecture changes temporally. How to keep the fly under constant sleep pressure needs much more effort than the classic sleep deprivation experiments. We hope in the future we can find a way to successfully expand our results to homeostasis regulation.

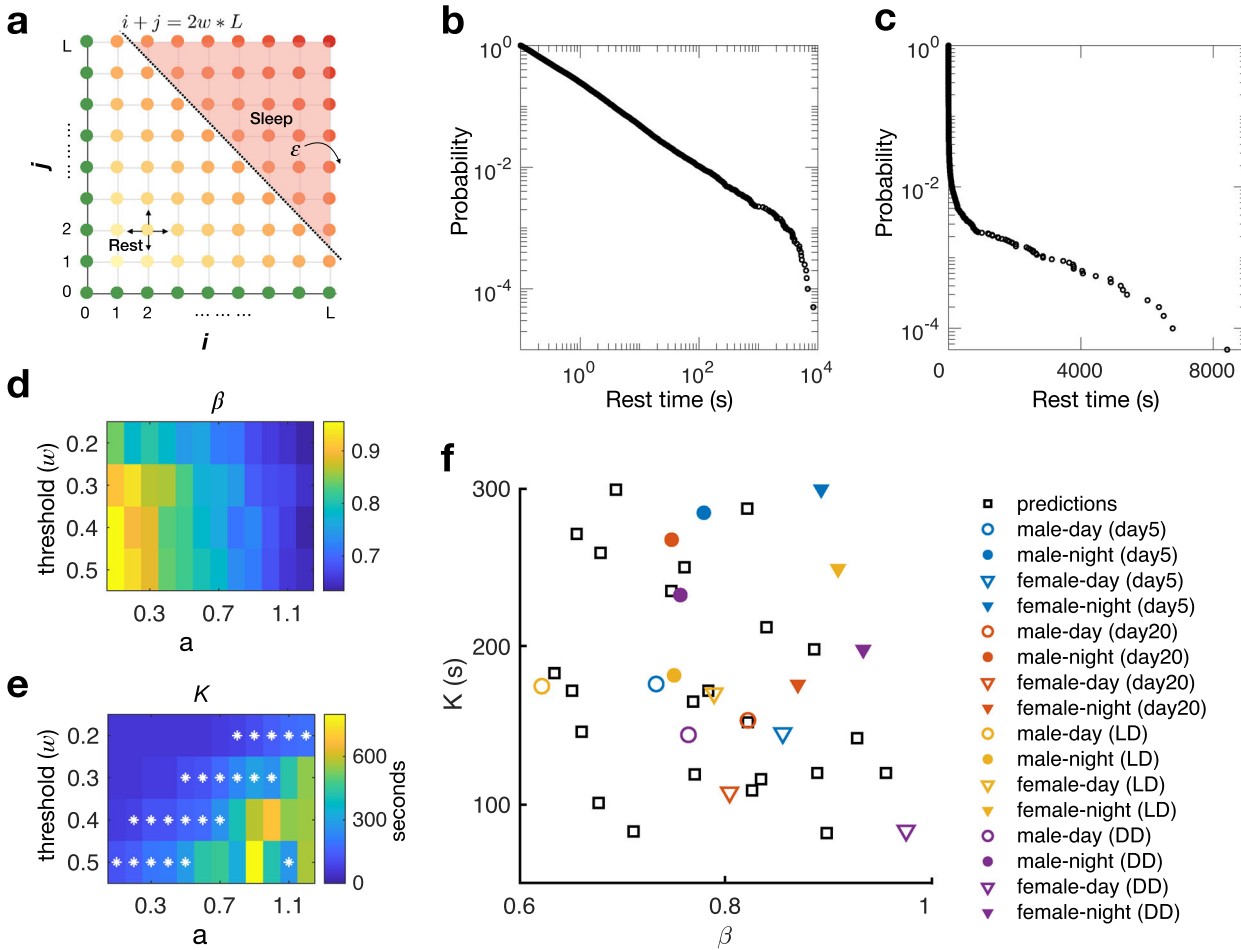

**Fig. 6 Mathematical model of sleep pattern. a** The schematic of the model. Assume each hemisphere has $L$ ($L = 50$ in our model) possible states when the fly is resting, the quietness of the hemisphere increases from state 1 to state $L$. The color of each state in the 2D space indicated how quiet the whole brain is. Fly would get back to motion if any of the hemispheres moves to state 0 (the green dots on the boundaries). If the brain state crosses the threshold line (black dotted line), the fly would enter the sleep phase (red dots) and would wake up with a constant rate $\varepsilon$. **b–c** Simulated results of the model. Statistics of the time the walker takes to get out of the $L \times L$ rest state space is similar to the distribution of the fly's rest time observed experimentally. **d** The active index generated by the model as a function of the parameters in the model. **e** The sleep latency generated by the model as a function of the parameters in the model. Asterisk: parameters that generated a sleep latency in the range of 80–300 s. **f** Comparison of the active index and sleep latency between those generated from the model with parameters marked by asterisks in (**e**) and those from the real data. Black squares represent the predicted values from the model. The experimental data is shown as the mean values of the flies from 4 scenarios (in 1D chamber with LD condition at day 5, in 1D chamber with LD condition at day 20, in 2D chamber with LD at day 5, and in 2D chamber with DD at day 5). Hollow circle, male in the light phase; filled circle, male in the dark phase; hollow triangle, female in the light phase; filled triangle, female in the dark phase.

Finally, it has not escaped our attention that our method can be used to analyze behaviors of mutant flies to study genes involved in different aspects of sleep at a level which was not previously possible.

## Methods

**Fly rearing conditions and behavior experiments**. All of the experiments were performed with the standard Drosophila laboratory strain, Canton-S. Flies were raised at 25°C (humidity: 60%) on standard yeast media with 12 h:12 h light: dark LED light cycle. Virgin flies were collected since they were born and grouped in plastic tubes. For all experiments, every single fly was put into the observing channel separately under short $CO_2$ anesthesia at day 3 post eclosion. Videos were taken after the flies had adapted to the environment for 24 hours. For the data of flies 5 days post eclosion was collected from days 5–7 for the following analysis. For the data of flies 20 days post eclosion, we raised the fly until 18 days and then collect data from days 20 to 22. For the constant darkness experiments, the flies were raised in the normal LD condition and then transferred to the DD condition at the same age as those in the other experiments.

The 1D chamber is 5 mm wide and 78 mm long, the food will occupy 15–20 mm of the length. The 2D chamber is an ellipse with 50 mm longer axes and 30 mm shorter axes. The depth of the Z-dimension of all the chambers is 2.5–3 mm. The fly walked on the ceiling sometimes, but most of the time it stayed or walked on the ground. Our observing chambers made of two parts, acrylic shaped base and glass plate above.

**Target tracking and rest behavior detection**. We simplified the image processing problem by segmenting the full images into partial images with only one fly in each chamber. We use difference images from the subtraction of the reconstructed background image to detect the location of the fly.

For each light phase or dark phase, we reconstructed the background picture by calculating the mean grey value of the first frames of each hour (total 12 frames). We can extract the area where the fly moves around by performing morphological opening on the greyscale background picture since the designed chambers were darker than the surroundings.

Only the difference of the pixels in the moving area was considered. Since the fly was the only moving object, we took the top 100 pixels that have the most difference. We then produced a binary image of the same size as the partial image with value 1 at the top 100 pixels. The largest connected component in this binary

image was found as where the fly was. The centroid of this largest connected area was used as the fly's location in the current frame. The trajectory of the fly is a time series [$x(t), y(t)$] of the locations, and the y-coordinate is referred to the longer axis of the 1D chamber or the 2D elliptic chamber.

The rest behavior is extracted from the trajectory. At each location of the trajectory, we calculated when the fly reached the location and when the fly left it. If the fly stayed at the same location for more than 1 second (25 frames), the fly had been resting at that location in that time interval.

To mimic the previous measurement method of circadian and sleep profile, DAMS, we used the midpoint of y-coordinate of the locations as an imaginary infrared beam. This midpoint divided the whole chamber into two parts. Every time the fly moved from one part to the other, it was regarded as crossing the infrared beam for once.

The raw videos were first transformed to avi format then processed with the Image Processing Toolbox provided by MATLAB 2019a.

**Estimation of parameters**. The rest events of each fly were classified by the light phase and dark phase. That means, we got the distributions of the rest time in the light phase and dark phase, respectively. The samples for the light phase distribution consisted of all the rest events when the light is on in the 3 shooting days, while the dark phase distribution consisted of all the rest events when the light is off in the 3 shooting days. In the paper, the data of the light phase was labeled as "day", and the data of the dark phase was labeled as "night".

We used the MATLAB packages written by Aaron Clauset[41] to estimate the exponent of power law distribution. The lower bound of the power law behavior was set as 3 s. We found the cutoff of power law distribution and the start of the exponential decay with one-sample Kolmogorov-Smirnov test. We scanned the $K$ value from 50 s. For each $K$, the rest samples longer than $K$ were subtracted by $K$ first and then tested with the null hypothesis that the data comes from an exponential distribution with the mean $\lambda$. The smallest $K$ that made the test fails to reject the null hypothesis at the 10% significance level was the sleep latency parameter $K$ for the individual fly, and the corresponding $\lambda$ was the sleep duration parameter.

The parameter $N$ indicates how many times the fly slept (was in rest longer than $K$) in 3 days in the experiment, which is also the total sample size of the exponential decay. We calculated the total sleep time of a fly as $N \times \lambda$. The total sleep time does not include the time when the fly was still in the sleep latency phase.

All the code for the data analysis and simulation is deposited and accessible on Zenodo[42].

**Statistics and reproducibility**. The two-sample $t$ test was used to determine whether the two data sets came from distributions with equal means. The samples assumed from populations with equal variances were analyzed by a two-tailed Student's $t$ test. The samples assumed with unequal variances were analyzed by two-tailed Welch's $t$ test. The test was performed with the embedded MATLAB function "$t$ test2" with 5% significance level of the hypothesis. The $p$ value of each test was indicated within the figures. Results were presented as mean ± standard error of mean (SEM). The total number (n) of individual flies for each experiment was indicated within the figures and in the legends. A minimum of three biologically independent repeats was used for each experiment. The flies were randomly sampled.

**Reporting summary**. Further information on research design is available in the Nature Research Reporting Summary linked to this article.

## Data availability
The datasets that support the findings of this study are available from the corresponding author upon reasonable request. The source data for the Figures and Supplementary Figures are provided with the paper as Supplementary Data 1.

## Code availability
All the code that supports the findings of this study is available at https://doi.org/10.5281/zenodo.4540736[42]

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

## Acknowledgements

We thank Kim Sneppen for discussion of the mathematical model, Xinyue Xia for assistance in data analysis, and Luhua Lai, Jingxiang Shen, Yongjun Qian, Shanshan Qin, Xiang Liu, Huixia Ren for discussions.

## Author contributions

X.X. and W.Y. designed the fly tracing platform and the behavior experiments. X.X., W.Y., B.T., X.S., and W.C. performed the behavior experiments. X.X. performed the data analysis. C.T. and Y.R. supervised the project. X.X. and C.T. wrote the paper. All authors reviewed and approved the paper.

## Competing interests

The authors declare no competing interests.
