## [Peer Review File · Communications Biology]

Reviewers' comments:

Reviewer #1 (Remarks to the Author):

Xu and Colleagues make a valid effort to provide alternative, more realistic approaches to quantifying sleep in *Drosophila*. Most of the field tracks activity of flies in small tubes, extracting sleep metrics based upon a 5-min inactivity threshold. There have been recent efforts to re-examine such data using probabilistic analysis (e.g., see Wiggin et al, 2020, PNAS). The current manuscript represents such an effort, albeit using somewhat different chambers and continuous video-monitoring rather than beam crossing. Like the Wiggin paper, they also find that probabilistic state-transition approaches provide perhaps better insight into sleep dynamic than bout length metrics alone. However, the manuscript is not ready for publication for a number of reasons. First, because this is a new approach employing different chambers and tracking methods, that has to be more fully validated. Ideally, a more thorough description of classic sleep metrics in these chambers, alongside heatmaps of fly positions over time, would be better prepare the reader for engaging in the subsequent probabilistic analyses, for comparison. For all we know, the flies might be walking on the ceiling, milling around near the food, escaping detection; we have no idea. The authors have to go through the trouble of better validating their new assay here. As stated by the authors, others have examined fractal stop/go statistics in flies, and theirs do not seem to align with published data. This is concerning, and suggests some aspect of behaviour in the chambers is different, hence another need for validation. Then, once validated, it does not appear sufficient to exemplify just five flies in Figure 2. Some summary stats seem to be needed here, alongside the individual examples. Next, having proposed a novel metric for identifying transitions from rest to sleep, they need to test the biological significance of this by performing some kind of sleep manipulation, to see if their metric (e.g., K) moves in a predictable direction. Having satisfied all of these necessary steps, only then does comparison with a model become appropriate. Before that, it seems a little premature. Tweaking the model (as Wiggin et al do, for example) should further validate observations made in the data with biological sleep manipulations. In summary, these results seem like an excellent new direction for *Drosophila* sleep research (the redesigned chambers and tracking system represents a necessary divorce from the extremely confined and unnatural context of Trikinetics tubes), but the manuscript unfortunately feels preliminary. The results herein would typically have constituted one methods figure for a longer, more thorough study centred on a biological question. I encourage the authors to pursue that, as they suggest in the last line of their discussion, and work their way towards the larger manuscript they are hoping for.

I've re-iterated some points below to help guide the authors toward an improved manuscript.

1. In order to show that these parameters are biologically relevant, the authors need to demonstrate some sort of manipulation that affects them. For example, do sleep deprived flies have a lower K value, since we would expect sleep onset to be more rapid. Do wake-promoting manipulations increase it?
2. The methods section lacks important information, including any information about the quantitative methods. For example:
 - How were the distributions determined from the empirical data? Were the distributions compared to alternatives using likelihood ratios?
 - Were the correlations performed in S4 corrected for multiple comparisons?
 - It's not clear if the data is from all days or an average per day? The description in the results says experiments were 4 days, but the methods section states 3 days.
3. Why is data only shown for 5 flies in Fig 2? This is especially important because you use this general trend in Fig4 to validate the model. Why isn't this averaged across the 34 flies?
4. Why is the data for female flies during the day not presented beyond S2? Presumably this dataset

exists for because the experiments weren't just conducted at night. Does it fit the model?

5. It is unclear where some of the data is derived from (perhaps due to the poorly explained methods), and some of it does not match what you would expect. For example total sleep time in Figure 3C is on the order of about 5-6 hours. That seems far too short across a 3 day sleep experiment.

6. The writing is difficult to follow in many sections, to the point where it was difficult to interpret the exact claim the authors were making about the data.

7. The terminology with respect to the current literature is confusing. For example, do the authors mean bout length when they mention 'sleep duration'?

8. The authors are using their own custom experimental setup for tracking fly sleep. I would like to see more validation of their approach to test if they see similar results (or not) for classical metrics in comparison to what others have shown in this type of open field setup.

- What are the chambers made of and what are their dimensions? Can flies walk on the ceiling of the chamber and how deep is the z dimension?

- Can you show a heat map of fly locations across the experiment? Is there bias, for example, to the end of the food or the side closest to the infrared illumination (i.e. from heat)?

9. The y-axis in Figure 2 doesn't make sense. It seems that this should be a count of the number of bouts or cumulative probability. These graphs are difficult to read because of the labelling on the x-axis.

Reviewer #2 (Remarks to the Author):

First of all, I am sorry to the authors and the editors for the delay of the review. I took a very long time due to the recent COVID-19 situation, which have affected largely our academic and personal activities.

In this manuscript, the authors described the analysis of temporal pattern of *Drosophila* locomotor activities using video recording data with high temporal and spatial resolution. This type of studies have been done in several laboratory, but the analysis is more profound and the conclusion dividing simple rest and sleep is interesting. I agree mostly with their discussion and recognize the significance of this work.

The major point I would like the authors to supplement to this work is quantitative statistical analyses. The most important conclusion of this paper is the different distribution pattern of rest and sleep. I recommend to use the method by Clauset et al (SIAM Review, 2009), which is adopted by the similar work by Ueno et al. (Ref 21).

In addition, the difficult point of power law distribution is the treatment and interpretation of its long tail and cutoff value (K). There are a limited number of data points in the long tail, which make the interpretation obscure. I would like the authors to discuss more in detail why they could conclude sleep does not consist the long tail of power law distribution of rest, but consists an exponential distribution, which is Gaussian decay pattern. It is better to refer previous studies distinguishing power law from exponential distributions.

Lastly, although the model the authors presented may be mathematically valid, it appears somewhat

contradictory to the conventional knowledge that sleep is regulated by homeostasis. I would like the authors to discuss how this model fit with homeostasis and how they speculate the biological mechanism of rest and sleep regulation.

Point-by-point response to reviewers' comments:

Reviewer #1:

Xu and Colleagues make a valid effort to provide alternative, more realistic approaches to quantifying sleep in *Drosophila*. Most of the field tracks activity of flies in small tubes, extracting sleep metrics based upon a 5-min inactivity threshold. There have been recent efforts to re-examine such data using probabilistic analysis (e.g., see Wiggin et al, 2020, PNAS). The current manuscript represents such an effort, albeit using somewhat different chambers and continuous video-monitoring rather than beam crossing. Like the Wiggin paper, they also find that probabilistic state-transition approaches provide perhaps better insight into sleep dynamic than bout length metrics alone. However, the manuscript is not ready for publication for a number of reasons. First, because this is a new approach employing different chambers and tracking methods, that has to be more fully validated. Ideally, a more thorough description of classic sleep metrics in these chambers, alongside heatmaps of fly positions over time, would be better prepare the reader for engaging in the subsequent probabilistic analyses, for comparison. For all we know, the flies might be walking on the ceiling, milling around near the food, escaping detection; we have no idea. The authors have to go through the trouble of better validating their new assay here. As stated by the authors, others have examined fractal stop/go statistics in flies, and theirs do not seem to align with published data. This is concerning, and suggests some aspect of behaviour in the chambers is different, hence another need for validation. Then, once validated, it does not appear sufficient to exemplify just five flies in Figure 2. Some summary stats seem to be needed here, alongside the individual examples. Next, having proposed a novel metric for identifying transitions from rest to sleep, they need to test the biological significance of this by performing some kind of sleep manipulation, to see if their metric (e.g., K) moves in a predictable direction. Having satisfied all of these necessary steps, only then does comparison with a model become appropriate. Before that, it seems a little premature. Tweaking the model (as Wiggin et al do, for example) should further validate observations made in the data with biological sleep manipulations. In summary, these results seem like an excellent new direction for *Drosophila* sleep research (the redesigned chambers and tracking system represents a necessary divorce from the extremely confined and unnatural context of Trikinetics tubes), but the manuscript unfortunately feels preliminary. The results herein would typically have constituted one methods figure for a longer, more thorough study centred on a biological question. I encourage the authors to pursue that, as they suggest in the last line of their discussion, and work their way towards the larger manuscript they are hoping for.

I've re-iterated some points below to help guide the authors toward an improved manuscript.

Response: We thank the reviewer for the professional and inspiring suggestions. Encouraged by the reviewer, we extended our study with careful validation of the experimental platform, comparison with the standard method, and discussion on the difference between the new method and the others. These changes are in multiple places in

the revised manuscript, but are especially reflected in Fig. 1, Fig. 2 and Fig. 4, Fig. S1-3, Discussion and Methods.

To compare our method with the classic ones, in the revised manuscript the measurements of the flies' circadian and sleep profiles with standard methods were shown alongside with the results from our method (Fig. 1C-F, Fig. 4A, B, D, E, and Fig. S4). The comparison indicates that the general sleep behavior of the flies in our experiments did not differ from that of the previous studies using classic methods. We even observed similar location preference in sleep (near the food source) as reported in others' experiments (Fig. S1E-I).

We analyzed if the shape and size of the chamber would make a difference on the sleeping pattern and found that the main conclusions did not change between 1D chamber (long tubular space) and larger sized 2D chamber (ellipse) (see Fig. S3; page 6 in the revised manuscript ("We note that the 1D chamber.....")).

We carefully compared our method with the one previously published in Wiggin et al, 2020, PNAS. We found these two methods are basically different from each other even though both methods explored the probability characteristics of the transition from rest to motion. We think our method is more straightforward and reflects the intrinsic laws of the sleep processes in fly. We have added a paragraph to discuss the situation of using hidden Markov model and how our method could explain the observed results by Wiggin's method (see Discussion on page 9, starting from "The Discrete Hidden Markov Model").

We have added the detailed information for the behavior experiments and data analysis in Methods section. The revised Methods (page11-12) included the chamber structure, target tracking, and parameter estimation. The information will be helpful if the readers would like to repeat or extend the experiments.

1. In order to show that these parameters are biologically relevant, the authors need to demonstrate some sort of manipulation that affects them. For example, do sleep deprived flies have a lower K value, since we would expect sleep onset to be more rapid. Do wake-promoting manipulations increase it?

Response: We very much agree with the reviewer's comment. In order to demonstrate the parameters are biologically relevant, we observed the sleeping pattern in older flies and under constant darkness condition (Fig. 4). We found the sleep phase parameters did change in different situations. More importantly, the changes of the sleep pattern were consistent with the conclusions from the previous studies while providing more accurate quantitative measurements. Thus, we think the parameters are biologically relevant. To further address this question, we have added a new section "Age-associated and light-associated changes in sleep pattern" starting from page 6 in the revised manuscript.

For the K value, we also observed difference from day to night, decreased in the older female flies and constant darkness. The K value in the sleep pattern represents the time scale when the fly's state transits from the power law decay to the exponential decay. It

resulted from the spontaneous activity of fly. It is hard to discuss the K simply in the sleep disturbing experiments since we can't keep the disturbing effect on the fly always the same (high constant sleep pressure). The homeostasis regulation happens all the time even though we know that the sleep architecture will change after temporal sleep disturbance. It may not be obvious how to define a statistically significant K value in a changing environment. We hope we can find a way to do this in the future. We have added a paragraph in Discussion (page 10) to address this question.

2. The methods section lacks important information, including any information about the quantitative methods. For example:

- How were the distributions determined from the empirical data? Were the distributions compared to alternatives using likelihood ratios?
- Were the correlations performed in S4 corrected for multiple comparisons?
- It's not clear if the data is from all days or an average per day? The description in the results says experiments were 4 days, but the methods section states 3 days.

Response: We are sorry for not explaining things clearly and have modified the Methods section. In Methods (under the Estimation of parameters) we now explained how we got the distribution for day and night. We compared the exponential model and the alternative gaussian model for fitting the sleep phase data and calculated the relative likelihood with AIC (Fig. S2). The exponential model showed its superiority in fitting the distribution from the day and night.

We have not corrected for the multiple comparisons. Here we only want to show the correlations between parameters.

We started shooting and collecting data after the flies had been in the chamber for 1 day. Thus, the total time the fly stayed in the chamber were 4 days, while the data were collected from the last 3 days. We modified this confusing text in the manuscript (first lines on page 3).

3. Why is data only shown for 5 flies in Fig 2? This is especially important because you use this general trend in Fig4 to validate the model. Why isn't this averaged across the 34 flies?

Response: In the revised manuscript, we presented at least 15 examples for male and female flies in the figures (Fig. 2) showed the distribution of samples merged from each repeat (Fig S3). As the reviewer suggested, we used the averaged values of each parameter from each condition in the comparison with the mathematical model in Fig. 6F (which is the original Fig. 4).

4. Why is the data for female flies during the day not presented beyond S2? Presumably this dataset exists for because the experiments weren't just conducted at night. Does it fit the model?

Response: In the revised manuscript, we presented the female's data and discussed about it along with male. It also fits the model (also see text on page 4 under the section "Quantitative sleep pattern of *Drosophila Melanogaster*").

5. It is unclear where some of the data is derived from (perhaps due to the poorly explained methods), and some of it does not match what you would expect. For example total sleep time in Figure 3C is on the order of about 5-6 hours. That seems far too short across a 3 day sleep experiment.

Response: We are sorry for the poorly explained methods. In the revised methods, the reviewer can find information about how we calculated the total sleep time in 3 days. The total sleeping time in our model excluded the resting time before the value K, so it would be comparable to the total amount of sleep time above the 5 min threshold if calculated with the standard method. That's probably why the reviewer felt it is too short. If we look at the sleep profile of the flies, it is similar to those in other studies. In the revised manuscript, in order to make it convenient to compare with the other studies, we showed the sleep profiles in a way mimicking the classic method (Fig. 3, Fig. 4, Fig. S4). The way calculating the total sleep was emphasized in Fig. 2E and in text on page 5 (right above the section "Parameters of sleep pattern").

6. The writing is difficult to follow in many sections, to the point where it was difficult to interpret the exact claim the authors were making about the data.

Response: We have tried to modify the manuscript to make it clearer.

7. The terminology with respect to the current literature is confusing. For example, do the authors mean bout length when they mention 'sleep duration'?

Response: We are sorry for this confusing term. We corrected it in the revised manuscript. Now the "sleep duration" in the text corresponded to λ , the mean length of time staying at the exponential decay sleep phase.

8. The authors are using their own custom experimental setup for tracking fly sleep. I would like to see more validation of their approach to test if they see similar results (or not) for classical metrics in comparison to what others have shown in this type of open field setup.

- What are the chambers made of and what are their dimensions? Can flies walk on the ceiling of the chamber and how deep is the z dimension?
- Can you show a heat map of fly locations across the experiment? Is there bias, for example, to the end of the food or the side closest to the infrared illumination (i.e. from heat)?

Response: We again thank the reviewer for the professional suggestion. In the revised methods, we add the information about the chamber (page 11). The 1D chamber is 5mm wide and 78mm long, the food will occupy 15-20mm of the length. The 2D chamber looks like an ellipse (see picture below) with longer axes 50mm and shorter axes 30mm. The depth of the z dimension of all the chambers is 2.5-3 mm. Sometimes we saw the fly walking on the ceiling, but most of the time, it stayed or walked on the ground. Our observing

chambers made of two parts, acrylic shaped base and glass plate above. We showed the heat map of fly locations during long-term rest. Mostly, the flies prefer to sleep near the food (Fig. S1E-I).

9. The y-axis in Figure 2 doesn't make sense. It seems that this should be a count of the number of bouts or cumulative probability. These graphs are difficult to read because of the labelling on the x-axis.

Response: We thank the reviewer for pointing this out. We corrected the y-axis label and now call the complementary cumulative probability as "probability" throughout the revised manuscript to avoid confusion. It is also explained in the manuscript (see text on page 4 under the section "Quantitative sleep pattern of *Drosophila Melanogaster*").

Reviewer #2:

First of all, I am sorry to the authors and the editors for the delay of the review. I took a very long time due to the recent COVID-19 situation, which have affected largely our academic and personal activities.

In this manuscript, the authors described the analysis of temporal pattern of *Drosophila* locomotor activities using video recording data with high temporal and spatial resolution. This type of studies have been done in several laboratory, but the analysis is more profound and the conclusion dividing simple rest and sleep is interesting. I agree mostly with their discussion and recognize the significance of this work.

The major point I would like the authors to supplement to this work is quantitative statistical analyses. The most important conclusion of this paper is the different distribution pattern of rest and sleep. I recommend to use the method by Clauset et al (SIAM Review, 2009), which is adopted by the similar work by Ueno et al. (Ref 21).

Response: We thank the reviewer for the inspiring and professional comments and suggestions. We completely understand the delay due to the COVID-19. In fact, this revision has been significantly delayed as well. Let's hope things will get back to normal soon.

We now used the methods by Clauset et al to estimate the exponent of the power law decay in the revised manuscript and explained how we estimated the parameters (see Methods/Estimation of parameters on page 12).

In addition, the difficult point of power law distribution is the treatment and interpretation of its long tail and cutoff value (K). There are a limited number of data points in the long tail, which make the interpretation obscure. I would like the authors to discuss more in detail why they could conclude sleep does not consist the long tail of power law distribution of rest, but consists an exponential distribution, which is Gaussian decay pattern. It is better to refer previous studies distinguishing power law from exponential distributions.

Response: We fully understand the reviewer's concern and compared the fitting of a Gaussian model with the exponential model (Fig. S2 and text on page 4). The Gaussian function decays much faster than the exponential decay since it decays as the $-x^2$ in the semi-logarithmic axes. The relative likelihood showed that it's more likely to be an exponential decay rather than a Gaussian decay.

By merging 3 days data together, we found that the enriched sample size made it possible to distinguish the long tail as exponential decay.

Lastly, although the model the authors presented may be mathematically valid, it appears somewhat contradictory to the conventional knowledge that sleep is regulated by homeostasis. I would like the authors to discuss how this model fit with homeostasis and how they speculate the biological mechanism of rest and sleep regulation.

Response: We agree with the reviewer that the manuscript lacks the discussion about the homeostasis of sleep regulation in our results. Sleep homeostasis is crucial for the individual to maintain proper sleep amounts after its sleep is disturbed. Here in our experiments, the flies slept and woke up spontaneously, so we regarded the sleep pressure as not changed much during the same light phase (sleep pressure have not been increased dramatically and only fluctuated slightly). We assumed that flies were at a similar state when they started to rest. This hypothesis is based on the observation that the current rest time did not depend on the total time of previous rest time (Fig. S6).

We have added some discussion about the homeostasis in the Mathematical model section (page 8) and Discussion section (page 10). Applying our model to analyze the homeostasis is possible since after sleep deprivation, the sleep architecture changes temporally. But how to keep the fly under a constant sleep pressure need much more effects than the classic sleep deprivation experiments. We hope in the future we can find a way to successfully expand our results to homeostasis regulation.

Reviewers' comments:

Reviewer #1 (Remarks to the Author):

The authors have made significant efforts to improve the manuscript. They have addressed my concerns about using a new tracking system and chambers for the behaviour by providing the heatmaps and comparing differences between 1D and 2D chambers. The pooled summaries for the examples shown in Figure 2 are now displayed in S3.

The addition of biological manipulations make the model far more convincing. There are a few results for the circadian data that may require further explanation:

- How can the reduction of male sleep be due to less bouts, when 'N' does not significantly decrease? Is this just a missing asterisk, as there is a reasonable difference in the histograms.
- In females, shorter sleep latency would mean falling asleep more quickly. How would this cause less sleep?
- In females, explain why is sleep duration and latency decreasing during the night, if flies are not sleeping less? This is different from the day result.

In general the writing is much clearer than the original manuscript. There is still a fair amount of awkward phrasing however. It would be worth having a third party to correct these before resubmission/publishing. Some examples (non-exhaustive):

- 33: study > studies
- 26: 'the mammalian organism'
- 53: of fruit fly
- 56: a different environment
- 71-73: awkward. 'been stayed'
- 86: parts
- 224: remind?

Other minor issues:

- The y-axis in Figure S1F-I should be changed to 'Average bout count per fly' for clarity.
- Make it clear earlier in the text when discussing the results in Figure 3 that this is in 1D chambers.
- It would be helpful in Figure 3E to add in the label that 'total' is $\lambda * N$.
- Please clarify 'control' in Figure 4. As I understand, in the aged experiments in the top panels this refers to younger flies, and in the bottom circadian experiments this refers to LD? These panels would be easier to understand using these terms instead of control. Additionally, labelling each metric with its own panel letter would make referring to it in the text much easier, and headings to separate out the top and bottom halves of this figure would improve its readability.

Reviewer #2 (Remarks to the Author):

The authors sincerely responded to my comments and revised the manuscript satisfactory. I have no more criticism.

Point-by-point response to reviewers' comments:

Reviewer #1:

The authors have made significant efforts to improve the manuscript. They have addressed my concerns about using a new tracking system and chambers for the behaviour by providing the heatmaps and comparing differences between 1D and 2D chambers. The pooled summaries for the examples shown in Figure 2 are now displayed in S3.

Response: We very much appreciate the reviewer's effort to help us improving the manuscript. In the second revision, we modified the figures, added more descriptions/explanations of the results, and corrected grammar errors, as detailed below.

The addition of biological manipulations make the model far more convincing. There are a few results for the circadian data that may require further explanation:

- How can the reduction of male sleep be due to less bouts, when 'N' does not significantly decrease? Is this just a missing asterisk, as there is a reasonable difference in the histograms.

Response: We agree with the reviewer that there is a reasonable difference in "N" of male in the day (new Fig. 4K). We did not put an asterisk there in the previous manuscript since we only labelled the test with an asterisk where the p-value < 0.05. We checked the p-value again and found it actually quite close to the significance level (p-value = 0.062). To avoid potentially misunderstanding for this and other figures, we have now labelled the p values directly in the figures instead of *, **, or ***. We have also mentioned that the male flies had fewer sleep bouts in the revised manuscript (line 244).

- In females, shorter sleep latency would mean falling asleep more quickly. How would this cause less sleep?

Response: In the standard method, the 5 minutes threshold is for the overall rest length (sum of sleep latency and sleep duration). The sleep profiles showed the females slept less under DD conditions, it may be due to shorter sleep latency, or shorter sleep duration, or both. In the new method, the Total sleep = $\lambda * N$, total sleep will decrease if only λ (sleep duration) decreases but the N does not increase. In our measurement, the females showed overall shorter rest and had both shorter sleep latency and short duration. It indicates the females fall asleep quickly but will maintain shorter sleep under DD.

- In females, explain why is sleep duration and latency decreasing during the night, if flies are not sleeping less? This is different from the day result.

Response: We totally agree with the reviewer's intuition. From the previous figure, we can see the height of the bar for the total sleep also decreased for the night. Here, we had the same problem as the first question. We did not make a strong conclusion about it since the p-value is 0.068. The sleep of females decreased at night indeed. The changes were small and hardly detected by the sleep profile measurements. With our method, we can quantify the differences directly through the lambda and K parameter. Especially, the decreases in nighttime sleep could be detected by lambda clearly. We modified the manuscript about this part (line250-258).

In light of these comments raised by the reviewer, we have added some more discussion/explanation on Fig. 4 (line 241-249)

In general the writing is much clearer than the original manuscript. There is still a fair amount of awkward phrasing however. It would be worth having a third party to correct these before resubmission/publishing. Some examples (non-exhaustive):

- 33: study > studies
- 26: 'the mammalian organism'
- 53: of fruit fly
- 56: a different environment
- 71-73: awkward. 'been stayed'
- 86: parts
- 224: remind?

Response: We have carefully gone over the manuscript to correct grammar and expression mistakes.

Other minor issues:

- The y-axis in Figure S1F-I should be changed to 'Average bout count per fly' for clarity.

Response: We changed the y-axis of Fig. S1F-I to "Average bout count per fly".

- Make it clear earlier in the text when discussing the results in Figure 3 that this is in 1D chambers.

Response: We added the phrase "In 1D chambers" in front of the sentence when discussing about Fig. 3 (line 173).

- It would be helpful in Figure 3E to add in the label that 'total' is $\lambda * N$.

Response: Now the "total" headings of figures have been changed to " $Total = \lambda * N$ ".

- Please clarify 'control' in Figure 4. As I understand, in the aged experiments in the top panels this refers to younger flies, and in the bottom circadian experiments this refers to LD? These panels would be easier to understand using these terms instead

of control. Additionally, labelling each metric with its own panel letter would make referring to it in the text much easier, and headings to separate out the top and bottom halves of this figure would improve its readability.

Response: Many thanks for the reviewer's suggestions. We modified the figures and manuscripts by specifically describing the group as "young flies", "5 days", "LD condition" et. al., instead of "control". The Fig. 4 is separated into two halves with titles, and each subfigure is labelled independently now.

REVIEWERS' COMMENTS:

Reviewer #1 (Remarks to the Author):

The authors have addressed all of my remaining concerns. This is a valuable study that helps move our field in the right direction, and I recommend publication. I appreciate the authors' patience and diligence in working towards improving their manuscript.